# Atrial Fibrillation Increases the Risk of Early-Onset Dementia in the General Population: Data from a Population-Based Cohort

**DOI:** 10.3390/jcm9113665

**Published:** 2020-11-14

**Authors:** Dongmin Kim, Pil-Sung Yang, Gregory Y.H. Lip, Boyoung Joung

**Affiliations:** 1Division of Cardiology, Department of Internal Medicine, College of Medicine, Dankook University, Cheonan-si, Chungnam 31116, Korea; kdongmin@dkuh.co.kr; 2Department of Medicine, The Graduate School, Yonsei University, Seoul 03722, Korea; 3Department of Cardiology, CHA Bundang Medical Centre, CHA University, Seongnam 13496, Korea; psyang01@cha.ac.kr; 4Liverpool Centre for Cardiovascular Science, University of Liverpool and Liverpool Heart & Chest Hospital, Liverpool L14 3PE, UK; 5Division of Cardiology, Department of Internal Medicine, Severance Cardiovascular Hospital, Yonsei University College of Medicine, Seoul 03722, Korea; cby6908@yuhs.ac

**Keywords:** atrial fibrillation, dementia, early-onset, prognosis, age

## Abstract

Atrial fibrillation (AF) is considered a risk factor for dementia, especially in the elderly. However, the association between the two diseases is not well identified in different age subgroups. The association of incident AF with the development of dementia was assessed from 1 January 2005, to 31 December 2013, in 428,262 participants from a longitudinal cohort (the Korea National Health Insurance Service-Health Screening cohort). In total, 10,983 participants were diagnosed with incident AF during the follow-up period. The incidence of dementia was 11.3 and 3.0 per 1000 person-years in the incident-AF and without-AF groups, respectively. After adjustment for clinical variables, the risk of dementia was significantly elevated by incident AF, with a hazard ratio (HR) of 1.98 (95% confidence interval [CI]: 1.80–2.17, *p* < 0.001), even after censoring for stroke (HR: 1.74, 95% CI: 1.55–1.94, *p* < 0.001). The HRs of incident AF for dementia onset before the age of 65 (early-onset dementia) and for onset after the age of 65 (late-onset dementia) were 2.91 (95% CI: 1.93–4.41) and 1.67 (95% CI: 1.49–1.87), respectively. Younger participants with AF were more prone to dementia development than older participants with AF (*p* for trend < 0.001). AF was associated with an increased risk of both early- and late-onset dementia, independent of clinical stroke.

## 1. Introduction

With the aging of the population, atrial fibrillation (AF) and dementia have become important health-care problems worldwide. AF is the most common sustained cardiac arrhythmia and is widespread in the older age group, leading to substantial public health and economic burdens [1,2]. Patients with AF have an increased risk of mortality and morbidity due to stroke, congestive heart failure, and hospitalization, in association with an increase in comorbid chronic diseases [3]. 

Approximately 40 million people worldwide have dementia, and this number is projected to increase owing to population aging [4]. Although the exact mechanism of the pathophysiology of dementia has not been well elucidated, a growing body of evidence suggests that AF might contribute to the development of cognitive dysfunction and dementia [5,6,7,8]. Oral anticoagulant therapy was associated with a reduced risk of developing dementia in an elderly AF population [8]. 

In contrast to the aged population, there is little evidence on the association of AF and early-onset dementia (EOD), with onset at age < 65 years. EOD can be a more devastating condition affecting patients who still have an active socioeconomic involvement. 

In this study, we investigated the association between incident AF and the risk of EOD and late-onset dementia (LOD) in a general population cohort including middle-aged groups, using the nationwide population-based database of the Korea National Health Insurance Service-Health Screening (NHIS-HEALS) [9].

## 2. Experimental Section

### 2.1. Source of Study Data

This study was based on Korea NHIS-HEALS data released in 2015 [9]. The dataset included 514,866 Koreans comprising a 10% random sample of all health-screened participants aged 40–79 years as an initial 2002–2003 cohort, who were followed-up through 2013 with data related to lifestyle and behavior from a questionnaire survey and the major results of health examinations. The NHIS-HEALS database incorporates sociodemographic information of the beneficiaries, a medical claims dataset including information on diagnosis based on the 10th revision of the International Classification of Disease (ICD-10) codes, medical bill details, medical treatments, disease histories, prescription drug use, and personal information of inpatients and outpatients in the National Health Information Database. The national health screening program is conducted biennially and includes regular blood tests, chest radiographic examinations, physical examination, and questionnaire survey on medical history. The date and cause of death of individual participants were extracted from the death registration database of the Korea National Statistical office and were linked to the NHIS-HEALS database. The Korean social security numbers linked every individual in the cohort, and all social security numbers were deleted after constructing the cohort by assigning serial numbers to prevent leakage of personal information. This study was approved by the Institutional Review Board of the Yonsei University Health System (Seoul, Republic of Korea, IRB No, 4-2020-0827), and the need for informed consent was waived. 

### 2.2. Study Population

From the Korean NHIS-HEALS database, a total of 457,510 individuals who underwent a health check-up between 2005 and 2010 were enrolled. Follow-up data were reviewed until December 2013. Participants with the following conditions were excluded: (i) valvular heart disease (presence of prosthetic heart valves, diagnosis of mitral stenosis, or insurance claims for valve replacement or valvuloplasty) (*n* = 1606), (ii) a history of ischemic stroke or transient ischemic attack before enrollment (*n* = 22,515), (iii) a history of hemorrhagic stroke before enrollment (*n* = 1130), (iv) a diagnosis of dementia before enrollment (*n* = 506), and (v) a diagnosis of AF before enrollment (*n* = 3491). Finally, we included 428,262 participants, among whom 10,983 had incident AF during the follow-up period (Figure 1).

The definition of AF was based on ICD-10 code I48. To ensure diagnostic accuracy, participants were considered to have AF only when AF was a discharge diagnosis or had been confirmed at least twice in the outpatient department. The positive predictive value of this definition of AF was 94.1% and was previously validated in the NHIS database [1,2,8].

### 2.3. Assessment of Dementia

Patients with dementia are registered in the national registry for severe disease and can receive medical expense reductions from the Korean government. Therefore, the assignment of the dementia diagnostic code is well controlled. The diagnosis of dementia was defined using the ICD-10 codes for dementia (F00 or G30 for Alzheimer’s dementia, F01 for vascular dementia, F02 for dementia with other diseases classified elsewhere, and F03 or G31 for unspecified dementia) with prescription of dementia drugs (rivastigmine, galantamine, memantine, or donepezil). To evaluate the accuracy of our definition of dementia, a validation study was conducted at two teaching hospitals with a total of 972 patients, using the medical records of the patients and the results of cognitive function tests. The positive predictive value was 94.7%.

### 2.4. Covariates

The medical claims and information on prescribed medication prior to the index date were used to define baseline comorbidities. When the condition was a discharge diagnosis or had been confirmed at least twice in an outpatient department, then the participants were considered to have comorbidities, as in previous studies with the NHIS (Appendix A) [1,2,8,10]. Baseline income status was estimated by the total amount of national health insurance premiums paid by the individual in the index year, proportional to the individual’s income.

### 2.5. Statistical Analysis

The baseline characteristics of the participants with and without incident AF were compared using logistic regression models, adjusting for age and sex where appropriate. The association between incident AF and incident dementia was assessed by Cox proportional hazard regression models. Incident AF was entered into the models as a time-varying variable. Follow-up began on the date of enrollment into the study and ended on the date of dementia diagnosis, date of death, or end of the study period (31 December 2013; defined as the last date of follow-up), whichever came first. In model I, we adjusted for age and sex. In model II, we adjusted for additional covariates, including hypertension, diabetes mellitus, dyslipidemia, previous myocardial infarction, peripheral artery disease, heart failure, chronic kidney disease, osteoporosis, chronic obstructive lung disease, liver disease, history of a malignant neoplasm, economic status, cardiovascular medications (aspirin, P2Y12 inhibitor, statin, anticoagulation agents, beta-blockers, angiotensin-converting enzyme inhibitors or angiotensin receptor blockers, calcium channel blockers, digoxin, diuretics), body mass index, blood glucose level, total cholesterol, and alcohol and smoking habits. Missing values were excluded case-wise. 

Stroke censoring was done for the sensitivity analyses. Participants were censored on the date of the stroke, if it developed before the end of the follow-up period. Any potential effect of age was assessed by stratifying the analysis at each decade of age and by using an interaction. Furthermore, propensity score matching was used to minimize potential systematic differences between the AF and the without-AF groups. The propensity scores for the incident AF of each study participant were calculated and adjusted for the above-mentioned variables in a logistic regression analysis. All tests were two-tailed, and *p* < 0.05 was considered significant. Statistical analyses were conducted with SAS version 9.4 (SAS Institute, Cary, NC, USA).

## 3. Results

### 3.1. Baseline Characteristics

Over a follow-up period of 310,6109 person-years, incident AF was diagnosed in 10,983 participants (3.5/1000 person-years). Participants with incident AF were older (age 61.7 ± 9.9 vs. 55.5 ± 9.1 years, *p* < 0.001), showed a male predominance, had more comorbidities, and were taking more cardiovascular medications. The follow-up duration was longer in the AF group (96 months, interquartile range (IQR) 86–101 months) than in the group without AF (93 months, IQR 84–100 months) (*p* < 0.001) (Table 1). At baseline, 0.44% of participants in the AF group received anticoagulation therapy. After diagnosis of AF, anticoagulation therapy was performed in 16.9% of participants in the AF group.

### 3.2. Risk of Dementia in the Overall Population

Compared to the group without AF, the cumulative incidence of dementia was higher in the incident-AF group (log-rank *p* < 0.001, Figure 2A). Among patients with incident AF, 880 (8.0%) participants developed dementia during 77,851 person-years of follow-up, compared with 9172 participants (2.2%) who developed dementia among patients without AF during 3,033,519 person-years. The incidence of dementia was 11.3 and 3.0 per 1000 person-years in the incident-AF and without-AF groups, respectively.

As quantified using the age- and sex-adjusted hazard ratios (HRs) with 95% confidence intervals (CIs), the risk of dementia was increased in participants with incident AF (HR: 2.06, 95% CI: 1.88–2.25). After additional adjustments for clinical variables, the risk of dementia was still significantly elevated by incident AF, with an HR of 1.98 (95% CI: 1.80–2.17) (Table 2, Figure 2A). Incident AF increased the risk of dementia in all decades of age. The risk of dementia due to AF was greater in young participants than in old participants (*p* for trend < 0.001) (Figure 3A). 

During the follow-up period, 12.7% and 2.0% of participants in the AF and without-AF groups developed stroke, respectively. After censoring for stroke, a higher cumulative incidence of dementia was observed in the incident-AF group than in the without-AF group (log-rank *p* < 0.001; Figure 2B). The incidence of dementia after censoring for stroke was 9.1 and 2.7 per 1000 person-years in the incident-AF and without-AF groups, respectively. Incident AF increased the risk of dementia, with an age- and sex-adjusted HR of 1.81 (95% CI: 1.63–2.02) and clinical variable-adjusted HR of 1.74 (95% CI: 1.55–1.94) (Table 2). After censoring for stroke, the trend of a greater risk of dementia due to AF in younger participants was still significant (*p* for trend < 0.001) (Figure 3B).

### 3.3. Risk of EOD and LOD

During the follow-up, EOD developed at a rate of 1.0 case per 1000 person-years and 0.3 cases per 1000 person-years in the AF and without-AF groups, respectively. Among EOD cases, Alzheimer’s dementia regarded 65.4% of the patients, whereas vascular and other types of dementia regarded 20.0% and 14.7% of the patients, respectively. AF showed a significantly higher association with EOD (clinical variable-adjusted HR: 3.81, 95% CI: 2.75–5.29) than with LOD (clinical variable-adjusted HR: 1.88, 95% CI: 1.70–2.07).

After censoring for stroke, incident AF increased the risk of both EOD and LOD, with clinical variable-adjusted HRs of 2.91 (95% CI: 1.93–4.41) and 1.67 (95% CI: 1.49–1.87), respectively (Table 2).

### 3.4. Risk of Dementia According to the Type of Dementia

Of all dementia cases, 73.7%, 13.0%, and 13.3% were classified as Alzheimer’s dementia, vascular dementia, and other dementia, respectively. In the AF group, Alzheimer’s dementia developed in 5.6% of participants, whereas 1.5% of patients in the without-AF group had Alzheimer’s dementia. The clinical variable-adjusted HRs for Alzheimer’s dementia were 1.69 (95% CI: 1.51–1.90) and 1.54 (95% CI: 1.35–1.76), including and censoring for stroke events during the observational period, respectively. The risk of vascular dementia was significantly high in the AF group (HR: 3.22, 95% CI: 2.61–3.99). After censoring for stroke, the HR for vascular dementia was still higher in the AF group than in the without-AF group (HR: 2.57, 95% CI: 1.95–3.38) (Table 2).

### 3.5. Subgroup Analyses 

The risk of dementia was significantly increased in all subgroups of the incident-AF group compared to the without-AF group, except in patients with chronic kidney disease and previous myocardial infarction (Figure 4).

### 3.6. Sensitivity Analysis with a Propensity Score-Matched Population

With propensity score matching, the baseline characteristics of the incident-AF and without-AF groups became similar (Appendix A). As quantified by age- and sex-adjusted HRs (95% CI), patients with incident AF had an elevated risk for dementia (HR: 1.90, 95% CI: 1.72–2.11). After additional clinical variable adjustments, the risk of dementia was still significantly elevated by incident AF, with an HR of 1.89 (95% CI: 1.70–2.09) (Appendix A). The increased risk for dementia of younger participants with incident AF was still significant (Appendix A).

The cumulative incidence of stroke-censored dementia was higher in the incident-AF group than the without-AF group (log-rank *p* < 0.001; Appendix A). The incidence of stroke-censored dementia per 1000 person-years was 9.1 and 6.4 in the incident-AF and without-AF groups, respectively. Incident AF increased the risk of dementia, with an age- and sex-adjusted HR of 1.68 (95% CI: 1.49–1.89) and a clinical variable-adjusted HR of 1.65 (95% CI: 1.46–1.86) (Appendix A). After censoring for stroke, incident AF elevated the risk of dementia in those older than 50 years (Appendix A). Both EOD and LOD were associated with AF. The greater risk of EOD than LOD due to AF was also observed in the propensity score-matched population (Appendix A).

## 4. Discussion

Our findings from this population-based study are consistent with those of previous studies, indicating that incident AF is associated with an elevated risk of dementia. In addition, we found this relationship was independent of clinical stroke and was significant in different age subgroups. AF was also associated with an elevated risk of EOD, and the risk of dementia due to AF was greater in younger participants. These results suggest a strong association between AF and dementia in various age groups. 

Previous studies have shown that the risk of dementia is associated with AF. However, various associations according to age were observed in different studies. In one study, AF increased the risk of dementia only in the younger age group [11]. In contrast, another study showed that only late-life AF was associated with dementia [12]. Previously, we reported similar results showing that AF increased the risk of dementia in an elderly population aged over 60 years from the Korean NIHS cohort [8]. In that study, AF was associated with all age groups. In the present study, we showed that the association was significant for middle- and late-life AF. The risk of dementia was higher in the younger age group.

Various mechanisms are suggested for the relationship between AF and dementia [13,14,15], including vascular mechanisms closely related to both Alzheimer’s disease and vascular dementia [16,17]. Stroke might also be a main mechanism of this association. The association between AF and dementia was stronger in individuals with both AF and stroke at baseline than in those without stroke at baseline. These findings support a possible interaction of stroke with AF and dementia. Although the association remained significant after censoring for stroke in our study, subclinical cerebral infarctions are known to be related to both AF and dementia [18]. 

Stroke is also known as a risk factor for EOD [19]. Various etiologies can result in presenile dementia, including genetic factors. Although genetic factors are considered strong risk factors for EOD in affected patients, familial dementia is extremely rare, and a very small proportion of EOD was attributable to familial dementia. Previous studies indicated that vascular risk factors including stroke might be more potent risk factors than genetic causes [19]. One study showed that stroke increased the risk of EOD by about three times. Another cross-sectional study showed a high prevalence of stroke or transient ischemic attack in the EOD group. In that study, other vascular risk factors, such as chronic kidney disease, ischemic heart disease, diabetes, hypertension, and peripheral vascular disease, were associated with EOD [20]. 

However, the effect of AF on EOD has not been well identified. In this study, incident AF was associated with an increased risk of dementia, even after censoring for stroke during follow-up and adjusting for vascular risk factors. Interestingly, while AF increased the risk of EOD by 3.88 times, it increased the risk of LOD by 1.88 times. Although we cannot provide the exact reason for this discrepancy, it might be explained by the causes of cerebrovascular events at a young age. Cardio-embolic stroke accounted for a significant proportion of the etiologies of stroke in the young-age group [21]. 

In terms of long-term effects of AF, a study proposed that the reason for the strong association between AF and dementia in the young-age group is the latent effect of AF, similar to those of other dementia risk factors, which also seems to differ with age [11]. Although the direct latent effect of AF on dementia could not be elucidated in this study, the vulnerability to dementia development was high in the young-age group, which supports the latent effect of AF.

Limitations of the present study were as follows. First, this study was based on administrative databases, and ICD-10 codes were used to define diseases. This type of study has potential vulnerabilities linked to inaccurate coding and definition. Appropriate validation of the definition is required to address these issues, and we have used the validated definition in previous studies based on a Korean NHIS cohort [1,2,8]. Dementia diagnosis was defined as in a previous study [22], and internal validation was performed for accuracy. Although we analyzed after censoring stroke, a significant proportion of participants with EOD and AF might already had subclinic or overt cerebral events, as stroke was defined by ICD-10 codes rather than brain imaging. Second, the type (paroxysmal vs. persistent) of AF could not be defined. As asymptomatic AF is not an uncommon condition, we may have misclassified some participants with this condition. Third, owing to the observational nature of this study, there might be residual unidentified confounders. Sleep apnea is an emerging risk factor for dementia and is also strongly associated with AF [23,24]. Unfortunately, we could not exclude possibilities that the relationship between AF and dementia was mediated by sleep apnea. To reduce those biases, we performed a sensitivity analysis with propensity score matching.

## 5. Conclusions

Incident AF was associated with an increased risk of both EOD and LOD, independent of clinical stroke. Younger participants with AF were more prone to dementia development than older participants with AF.

## Figures and Tables

**Figure 1 jcm-09-03665-f001:**
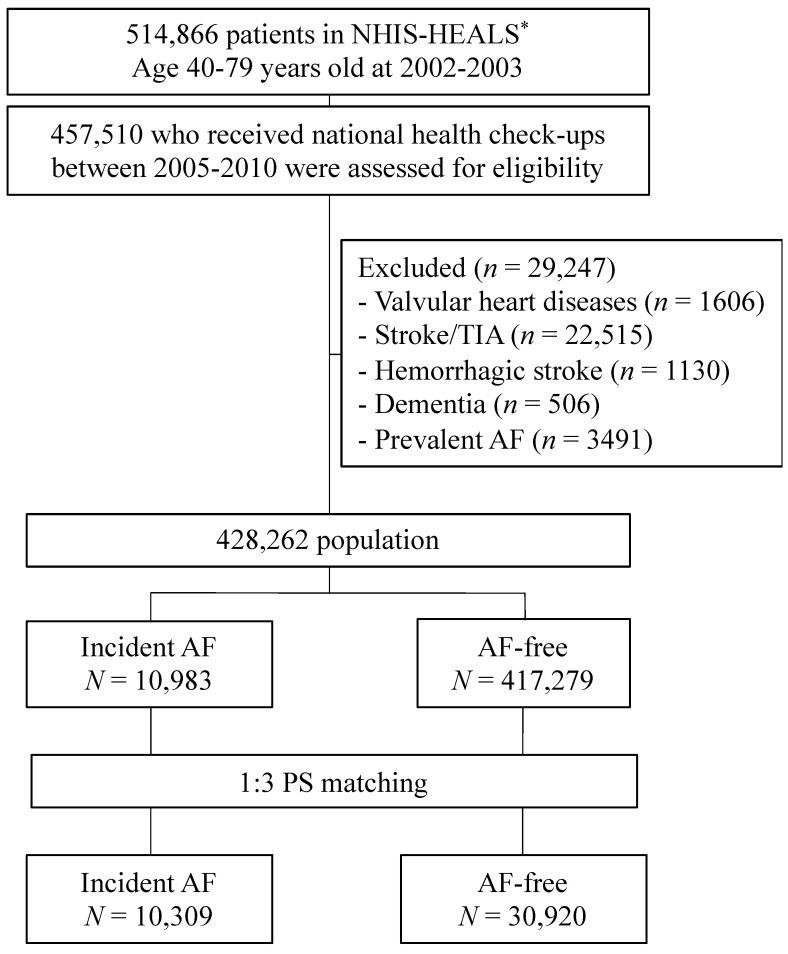
Flowchart of the enrollment procedure of the study population and analyses. TIA, transient ischemic attack; AF, atrial fibrillation. * The National Health Insurance Service-Health Screening Cohort (NHIS-HEALS) in Korea.

**Figure 2 jcm-09-03665-f002:**
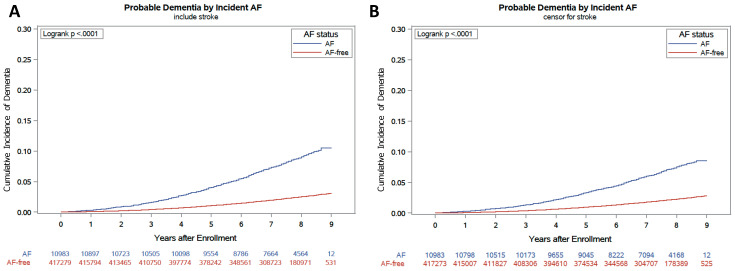
Cumulative incidence of dementia before (**A**) and after (**B**) censoring for stroke.

**Figure 3 jcm-09-03665-f003:**
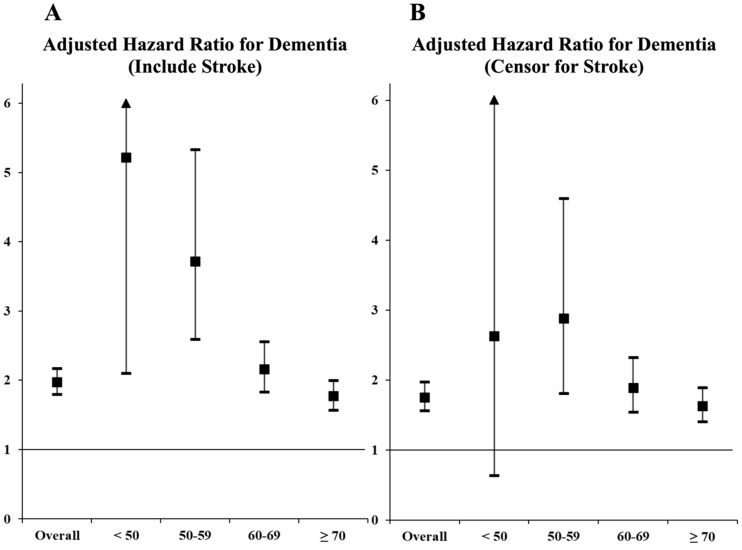
Hazard ratios for dementia per decade of age in the presence of AF. (**A**) Including stroke during the follow-up; (**B**) excluding stroke during the follow-up. The hazard ratios are expressed as boxes, the 95% confidence intervals are expressed as limit lines, and the horizontal line (at hazard ratio 1) indicates no difference in hazard ratios between the AF and the without-AF groups.

**Figure 4 jcm-09-03665-f004:**
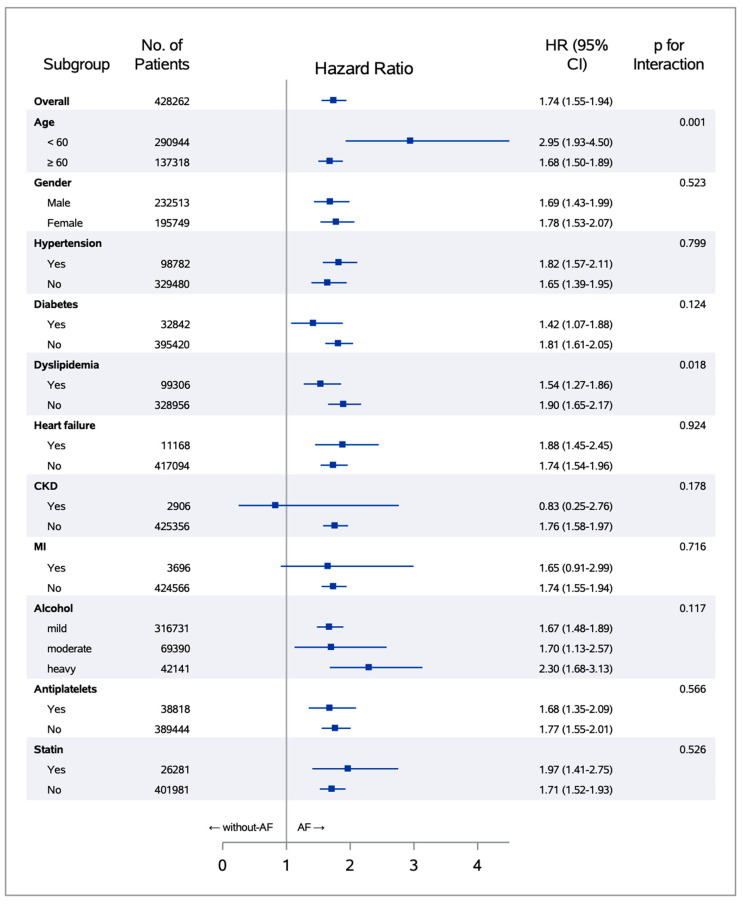
Subgroup analysis for the risk of dementia according to the AF status. The hazard ratios and the 95% confidence intervals are expressed as boxes and limit lines, respectively. The vertical line (at hazard ratio 1) indicates no difference in hazard ratios between AF and without-AF groups.

**Table 1 jcm-09-03665-t001:** Baseline characteristics.

	No AF	Incident AF	*p*-Value
	(*N* = 417,279)	(*N* = 10,983)	
Age, mean (SD), years	55.5 ± 9.1	61.7 ± 9.9	<0.001
Age > 65 No. (%)	74,143 (17.8)	4531 (41.3)	<0.001
Female, No. (%)	191,431 (45.9)	4318 (39.3)	0.002
BMI, mean (SD), kg/m^2^	24.0 ± 2.9	24.3 ± 3.1	<0.001
SBP, mean (SD), mmHg	125.8 ± 16.6	130.2 ± 17.6	0.072
DBP, mean (SD), mmHg	78.4 ± 10.7	80.1 ± 11.0	0.038
Blood glucose, mean (SD), mg/dL	98.4 ± 26.8	101.9 ± 32.5	0.008
Total cholesterol, mean (SD), mg/dL	198.9 ± 36.9	195.4 ± 38.0	<0.001
Serum creatinine, mean (SD), mg/dL	0.99 ± 0.97	1.08 ± 1.02	0.75
Blood hemoglobin, mean (SD), mg/dL	13.9 ± 1.5	13.9 ± 1.5	0.012
Hypertension, No. (%)	94,332 (22.6)	4450 (40.5)	<0.001
Diabetes, No. (%)	31,483 (7.5)	1359 (12.4)	0.003
Dyslipidemia, No. (%)	95,735 (22.9)	3571 (32.5)	<0.001
Heart failure, No. (%)	10,191 (2.4)	977 (8.9)	<0.001
History of MI, No. (%)	3423 (0.8)	273 (2.5)	0.207
PAOD, No. (%)	7150 (1.7)	356 (3.2)	0.809
CKD or ESRD, No. (%)	2764 (0.7)	142 (1.3)	0.032
Osteoporosis, No. (%)	56,584 (13.6)	2042 (18.6)	<0.001
COPD, No. (%)	10,609 (2.5)	708 (6.5)	<0.001
History of liver disease, No. (%)	84,708 (20.3)	2828 (25.8)	<0.001
History of malignancy, No. (%)	27,063 (6.5)	1118 (10.2)	<0.001
CHA_2_DS_2_-VASc score	1.0 ± 1.0	1.6 ± 1.3	<0.001
Income level			0.095
Low, No. (%)	120,422 (28.9)	3354 (30.5)	
Middle, No. (%)	151,912 (36.4)	3868 (35.2)	
High, No. (%)	144,945 (34.7)	3761 (34.2)	
Smoking			0.571
No, No. (%)	283,087 (71.7)	7384 (71.4)	
Former, No. (%)	36,450 (9.2)	1060 (10.3)	
Current, No. (%)	75,477 (19.1)	1894 (18.3)	
Alcohol consumption			0.004
Low, No. (%)	308,731 (74.0)	8000 (72.8)	
Moderate, No. (%)	67,780 (16.2)	1610 (14.7)	
Heavy, No. (%)	40,768 (9.8)	1373 (12.5)	
Exercise			<0.001
None, No. (%)	7882 (1.9)	153 (1.4)	
Seldom, No. (%)	10,850 (2.6)	158 (1.4)	
Regular, No. (%)	398,547 (95.5)	10,672 (97.2)	
ACE inhibitor or ARB, No. (%)	39,615 (9.5)	1982 (18.1)	0.313
Beta-Blocker, No. (%)	37,528 (9.0)	2015 (18.4)	<0.001
Diuretics, No. (%)	44,803 (10.4)	2294 (20.9)	0.073
Potassium-sparing diuretics, No. (%)	3791 (0.9)	305 (2.8)	0.361
NDHP-CCB, No. (%)	3744 (0.9)	340 (3.1)	<0.001
DHP-CCB, No. (%)	53,735 (12.9)	2478 (22.6)	0.16
Digoxin, No. (%)	889 (0.2)	249 (2.3)	<0.001
Alpha-blocker, No. (%)	8574 (2.1)	490 (4.5)	0.805
Statin, No. (%)	25,263 (6.1)	1018 (9.3)	<0.001
Antiarrhythmic agents, No. (%)	139 (0.03)	46 (0.42)	<0.001
Aspirin, No. (%)	36,385 (8.7)	2034 (18.5)	<0.001
P_2_Y_12_ inhibitor, No. (%)	2020 (0.5)	140 (1.3)	0.577
Anticoagulation, No. (%)	179 (0.04)	48 (0.44)	<0.001
F/U duration, median (IQR), months	93 (84, 100)	96 (86, 101)	<0.001

Values are expressed in No. (%), mean ± standard deviation (SD), or median (interquartile range; IQR). Abbreviation: BMI, body mass index; SBP, systolic blood pressure; DBP, diastolic blood pressure; MI, myocardial infarction; PAOD, peripheral artery occlusive disease; CKD, chronic kidney disease; ESRD, end-stage renal disease; COPD, chronic obstructive pulmonary disease; ACE, angiotensin-converting enzyme; ARB, angiotensin type II receptor blocker; NDHP, non-dihydropyridine; CCB, calcium channel blocker; DHP, dihydropyridine; F/U, follow-up; CHA2DS2-VASc score (congestive heart failure, blood pressure consistently above 140/90 mm Hg or treated hypertension on medication, age ≥ 75 years, diabetes mellitus, prior stroke, transient ischemic attack, or thromboembolism)–(vascular disease (e.g., peripheral artery disease, myocardial infarction, aortic plaque), age 65–74 years, female sex).

**Table 2 jcm-09-03665-t002:** Incidence of dementia during follow-up periods according to the AF status.

Dementia	Cases, No. (%)	Incidence *	Adjusted HR (95% CI)	
			Model I ^†^	Model II ^‡^
Including Stroke				
Overall dementia				
No AF (*n* = 417,279)	9172 (2.2)	3.0	1.00 (Reference)	1.00 (Reference)
AF (*n* = 10,983)	880 (8.0)	11.3	2.06 (1.88–2.25)	1.98 (1.80–2.17)
Early-onset dementia				
No AF (*n* = 409,145)	1038 (0.3)	0.3	1.00	1.00
AF (*n* = 10,177)	74 (0.7)	1.0	3.93 (2.88–5.38)	3.81 (2.75–5.29)
Late-onset dementia				
No AF (*n* = 416,241)	8134 (2.0)	2.7	1.00	1.00
AF (*n* = 10,909)	806 (7.4)	10.4	1.95 (1.78–2.15)	1.88 (1.70–2.07)
Alzheimer Dementia				
No AF (*n* = 417,253)	6797 (1.6)	2.2	1.00	1.00
AF (*n* = 10,981)	611 (5.6)	7.8	1.76 (1.58–1.97)	1.69 (1.51–1.90)
Vascular Dementia				
No AF (*n* = 417,253)	1159 (0.3)	0.4	1.00	1.00
AF (*n* = 10,981)	151 (1.4)	1.9	3.45 (2.81–4.24)	3.22 (2.61–3.99)
Censored for Stroke				
Overall dementia				
No AF (*n* = 417,279)	8246 (2.0)	2.7	1.00	1.00
AF (*n* = 10,983)	680 (6.2)	9.1	1.81 (1.63–2.02)	1.74 (1.55–1.94)
Early-onset dementia				
No AF (*n* = 409,145)	902 (0.2)	0.3	1.00	1.00
AF (*n* = 10,177)	53 (0.5)	0.7	2.92 (1.95–4.36)	2.91 (1.93–4.41)
Late-onset dementia				
No AF (*n* = 416,241)	7344 (1.8)	2.4	1.00	1.00
AF (*n* = 10,909)	627 (5.7)	8.4	1.75 (1.57–1.95)	1.67 (1.49–1.87)
Alzheimer Dementia				
No AF (*n* = 417,253)	6246 (1.5)	2.1	1.00	1.00
AF (*n* = 10,981)	493 (4.5)	6.6	1.61 (1.41–1.82)	1.54 (1.35–1.76)
Vascular Dementia				
No AF (*n* = 417,253)	911 (0.2)	0.3	1.00	1.00
AF (*n* = 10,981)	91 (0.8)	1.2	2.82 (2.16–3.67)	2.57 (1.95–3.38)

HR, hazard ratio. * Incidence: per 1000 person-year † Model I was adjusted for age and sex. ‡ Model II was additionally adjusted for hypertension, diabetes mellitus, dyslipidemia, heart failure, previous myocardial infarction, peripheral artery disease, osteoporosis, chronic kidney disease, chronic obstructive pulmonary disease, malignant neoplasm, liver disease, CHA2DS2-VASc score, cardiovascular medications (e.g., ACE inhibitors or ARB, beta-blockers, diuretics, statin, alpha-blockers, K-sparing diuretics, digoxin, calcium channel blockers, antiarrhythmic drugs, aspirin, P2Y12 inhibitors, oral anticoagulants), economic status, alcohol consumption, smoking status, exercise habits, follow-up duration, body mass index, systolic and diastolic blood pressure, blood glucose, total cholesterol, and blood hemoglobin level.

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
