# Peer review of "Atrial Fibrillation Increases the Risk of Early-Onset Dementia in the General Population: Data from a Population-Based Cohort"

_jcm, 2020, doi:10.3390/jcm9113665_

Round 1
Reviewer 1 Report
932304
Atrial fibrillation increases the risk of early-onset dementia in the general population: Data from a population-based cohort
Atrial fibrillation (AF) is considered a risk factor for dementia, especially in the elderly. However, the association between the two diseases is not well identified in different age subgroups
In this study, the authors investigated the associations between incident AF and the risk of early onset dementia (EOD) and late- onset dementia (LOD) in a general population cohort including middle-aged groups, using the 50 nationwide population-based database of the Korea National Health Insurance Service-Health 51 Screening (NHIS-HEALS)
Incident AF was associated with an increased risk of both EOD and LOD, independent of stroke. Younger participants with AF were more prone to dementia development than older participants with AF.
I have the following concerns:
-Methods: could the difference be mediated by sleep disturbrances? For example, a respiratory disorder such as OSAS is more dangerous at a young age due to the lack of preconditioning. However, considering the lack of these data, I suggest to add sleep disordered in exclusion criteria.
(add: Buratti L, Viticchi G, Baldinelli S, Falsetti L, Luzzi S, Pulcini A, Petrelli C, Provinciali L, Silvestrini M. Sleep Apnea, Cognitive Profile, and VascularChanges: An IntriguingRelationship. Journal of Alzheimer’sDisease2017;60:1195-1203)
Considering the relation between dementia and vascular disease:
.Viticchi G, Falsetti L, Buratti L, Sajeva G, Luzzi S, Bartolini M, Provinciali. L, Silvestrini M. Framingham Risk Score and the Risk of Progression from Mild Cognitive Impairment to Dementia. Journal of Alzheimer’s Disease 2017;59:67-75.
- 31. Falsetti L, Viticchi G, Buratti L, Grigioni F, Capucci A, Silvestrini M. Interactions between Atrial Fibrillation, Cardiovascular Risk Factors, and ApoE Genotype in Promoting Cognitive Decline in Patients with Alzheimer's Disease: A Prospective Cohort Study. J Alzheimers Dis. 2018;62(2):713-725. doi: 10.3233/JAD-170544.
Author Response
In this study, the authors investigated the associations between incident AF and the risk of early onset dementia (EOD) and late- onset dementia (LOD) in a general population cohort including middle-aged groups, using the nationwide population-based database of the Korea National Health Insurance Service-Health Screening (NHIS-HEALS). Incident AF was associated with an increased risk of both EOD and LOD, independent of stroke. Younger participants with AF were more prone to dementia development than older participants with AF.
Point 1: I have the following concerns:
-Methods: could the difference be mediated by sleep disturbrances? For example, a respiratory disorder such as OSAS is more dangerous at a young age due to the lack of preconditioning. However, considering the lack of these data, I suggest to add sleep disordered in exclusion criteria.
(add: Buratti L, Viticchi G, Baldinelli S, Falsetti L, Luzzi S, Pulcini A, Petrelli C, Provinciali L, Silvestrini M. Sleep Apnea, Cognitive Profile, and VascularChanges: An IntriguingRelationship. Journal of Alzheimer’sDisease2017; 60:1195-1203)
Response 1: Thank you. We agree that OSAS can be a possible mediator of these relationship. Unfortunately, we did not have any data on obstructive sleep apnea. In response to the reviewer’s comment, we added to the study limitation as follows; Sleep apnea is an emerging risk factor for dementia and is also strongly associated with AF. Unfortunately, we could not exclude possibilities that the relationship between AF and dementia was mediated by sleep apnea. (Page 11; line 283-285). We also citated following article; Journal of Alzheimer’s Disease2017; 60:1195-1203
Point 2: Considering the relation between dementia and vascular disease:
.Viticchi G, Falsetti L, Buratti L, Sajeva G, Luzzi S, Bartolini M, Provinciali. L, Silvestrini M. Framingham Risk Score and the Risk of Progression from Mild Cognitive Impairment to Dementia. Journal of Alzheimer’s Disease 2017;59:67-75.
- Falsetti L, Viticchi G, Buratti L, Grigioni F, Capucci A, Silvestrini M. Interactions between Atrial Fibrillation, Cardiovascular Risk Factors, and ApoE Genotype in Promoting Cognitive Decline in Patients with Alzheimer's Disease: A Prospective Cohort Study. J Alzheimers Dis. 2018;62(2):713-725. doi: 10.3233/JAD-170544. 

Response 2: We also agree with the reviewer’s comment that vascular diseases and risk factors for vascular disease are associated with dementia. In the revised manuscript, we discussed it intensively in the discussion section as follows; Various mechanisms suggested to postulate the relationship between AF and dementia [13-15]. Of those, vascular mechanisms closely related with both Alzheimer's disease and vascular dementia [16,17]. (Page 10; line 241-243)

Reviewer 2 Report
Although the findings are not new, the authors provide a well-designed study based on an administrative database with an effort to overcome inherent bias associated with such studies. Their conclusions add to the evidence and provide some insight into the relationship between AF and dementia.
Author Response
Point 1: Although the findings are not new, the authors provide a well-designed study based on an administrative database with an effort to overcome inherent bias associated with such studies. Their conclusions add to the evidence and provide some insight into the relationship between AF and dementia.
Response 1: Thank you. We agree with you. Previous studies showed that the relationship between AF and dementia was different according to age subgroups. However, we think that our findings, especially the association of AF and early-onset dementia, highlight that importance of prevention of AF and early aggressive treatment AF.

Reviewer 3 Report
In this article, Kim et al use a large registry to describe an increased incidence of dementia amongst subjects with coexisting AF compared to those without. This remained an independent risk for dementia after taking into account prior stroke, although the increased risk was significantly lower after censoring stroke. The risk was more pronounced in patients under age 65. The hazard ratio for dementia with incident AF was 1.98 after controlling for covariates.
Overall the paper is clearly written, and its findings are relevant.
The following are some minor suggestions:
- It is interesting that only a very small proportion of patients with incident AF were anticoagulated (48, 0.44%). It would be worthwhile if the authors added some discussion around this point, as it may be a major reason behind the increased dementia risk in those with AF, which is potentially preventable.
- It is highly likely, as alluded to by the authors, that silent stroke accounted for a significant proportion of patients with EOD with coexistent AF, even though clinical stroke was not recorded. This should be included in the limitations section.
Author Response
Point 1: It is interesting that only a very small proportion of patients with incident AF were anticoagulated (48, 0.44%). It would be worthwhile if the authors added some discussion around this point, as it may be a major reason behind the increased dementia risk in those with AF, which is potentially preventable.
Response 1: Thank you for comment. Incident AF was defined with ICD-10 code during observation periods. Defined AF group had diagnosis of AF after enrolment. We thought that participants were received their anticoagulation therapy due to other reasons rather than AF at the time of enrolment. Therefore, small fraction of participants was on anticoagulation in AF group as in the Table 1. After diagnosis of AF, 16.9% of patients had anticoagulation therapy. And in our preliminary analysis, anticoagulation reduced the incidence of dementia in patients with AF with variable-adjusted HR of 0.64 (95% CI: 0.49-0.84, p<0.001) (Figure below). We clarified this in results as follows; At baseline, 0.44% of participants had anticoagulation therapy in the AF group. After diagnosis of AF, anticoagulation therapy was performed in 16.9% of participants in the AF group. (Page 4; line 135-137)
Figure. Probable dementia according to anticoagulation status in participants with AF.
Point 2: It is highly likely, as alluded to by the authors, that silent stroke accounted for a significant proportion of patients with EOD with coexistent AF, even though clinical stroke was not recorded. This should be included in the limitations section.
Response 2: We thank the reviewer for the insightful comment. In response to the reviewer’s comment, we added to the limitations sections as following; Although we analyzed after censoring stroke, significant proportion of participants with EOD and AF might already had subclinic or overt cerebral events as stroke was defined by ICD-10 codes rather than brain imaging. (Page 11; line 279-281)
